# Herbal Polyphenols as Selenium Reducers in the Green Synthesis of Selenium Nanoparticles: Antibacterial and Antioxidant Capabilities of the Obtained SeNPs

**DOI:** 10.3390/molecules29081686

**Published:** 2024-04-09

**Authors:** Aleksandra Sentkowska, Julia Konarska, Jakub Szmytke, Anna Grudniak

**Affiliations:** 1Heavy Ion Laboratory, University of Warsaw, Pasteura 5A, 02-093 Warsaw, Poland; 2Department of Bacterial Genetics, Institute of Microbiology, Faculty of Biology, University of Warsaw, Miecznikowa 1, 02-096 Warsaw, Polandj.szmytke@student.uw.edu.pl (J.S.); a.grudniak@uw.edu.pl (A.G.)

**Keywords:** selenium nanoparticles, antioxidant activity, antibacterial properties, synthesis

## Abstract

Selenium is an essential trace element for the proper functioning of the human body. In recent years, great attention has been paid to selenium nanoparticles (SeNPs) due to their potential for medicinal applications. In this study, herbal extracts were used in the green synthesis of SeNPs. The influence of herbal species, the ratio of the reagents, and post-reaction heating on the antibacterial and antioxidant properties of obtained SeNPs were investigated. The relationship between these properties and the physical parameters of obtained nanoparticles (e.g., size, shape) was also studied. It has been proven that SeNPs showed higher antioxidant and antibacterial properties in comparison to herbal extracts taken for their synthesis. Heating of the post-reaction mixture did not affect the SeNP size, shape, or other studied properties.

## 1. Introduction

Nanotechnology is a promising new field of science with great potential that can be used in medicine. The development of nanomedicine may revolutionize the way we diagnose and treat many diseases, from infections with antibiotic-resistant bacteria to cancer [1]. One group of nanoparticles that has been intensively studied recently due to their potential medical applications is selenium nanoparticles (SeNPs). It was proven that SeNPs are less toxic than inorganic and organic forms of selenium [2]. More attention is also placed on studying their strong antioxidant and antibacterial effects [3,4]. Therefore, it is not surprising that many methods for their synthesis have been described in the literature. However, it seems that green methods are starting to replace the classic chemical approaches to selenium nanoparticle synthesis [5,6,7]. Of all the green synthesis methods, the most environmentally friendly are those that use plant extracts. This approach requires nontoxic solvents, mild temperatures, and the application of reducing agents that are easily accessible, cheap, biodegradable, and not harmful to the environment. Furthermore, it is not necessary to isolate microorganisms from the post-reaction mixture, which significantly increases the cost of the entire procedure, as well as makes it difficult to transfer to an industrial scale [8]. The main difficulty of this variant of SeNP synthesis is the limited control of the synthesis conditions, resulting from the very specificity of the used extract. The green synthesis approach via plant extract involves many secondary metabolites such as flavonoids, alkaloids saponins, carbohydrates, proteins, tannins, and steroids as natural reducers and/or stabilizers. For example, the content of polyphenols in tea leaves varies with climate, season, horticultural practices, and leaf age [9]. Therefore, repeating the analytical procedure, even using an extract from the same plant but from a different culture may be difficult. Another problem with the literature data is that the authors do not specify what factors determine the selection of a particular plant for SeNP synthesis. Some of them use little-known plants found only locally, which are, therefore, difficult to obtain by other researchers. In this study, well-known medicinal herbs such as lemon balm (*Melissa officinalis*), raspberry (*Rubus idaeus*), sage (*Salvia officinalis*), blackberry (*Rubus plicatus*), and hop (*Humulus*) were used. Their infusions have been used in folk medicine for many centuries, and knowledge development has allowed scientific confirmation of their effects. Lemon balm is used for its antibacterial, antifungal, antiparasitic, and fever-reducing properties, and as a therapeutic aid in the treatment of chronic headaches, indigestion, nausea, and insomnia [10]. Hops have been used for ages to treat gastric problems, toothaches, inflammation, and fever; they also serve as a preservative and are used in cattle feed [11]. The sedative and sleep-inducing properties of the plant have led to its current use in phytotherapy for the treatment of anxiety and sleep disorders [12]. However, the main use of hops remains in the beer industry, where it is an additive that gives the beverage a bitter taste. Raspberry leaves are commonly used in the treatment of flu, diabetes, fever, menstrual pain, and colic; when used externally, they have antibacterial and anti-inflammatory effects [13]. Sage is used in the treatment of, above all, inflammation of the skin and throat, hyperglycemia, gastric problems, or excessive sweating [14]. Both fruits and sage leaves are used in digestive disorders, act as hypoglycemic agents, and alleviate inflammation of the mucous membrane of the mouth and throat. The compounds contained in the plant also have an antiproliferative effect on cancer cells [15]. The use of such plant material allows the described procedures to be easily repeated by other research teams anywhere in the world. In the work, the content of individual polyphenolic compounds was determined using high-performance liquid chromatography coupled with mass spectrometry. To increase the sensitivity of the assay method, chromatographic separation was carried out in the HILIC mode. SeNPs are known for their antioxidant properties, which have been evaluated in the extracts themselves and in nanoparticle suspensions, using four different methods: the Folin–Ciocalteu method (FC), scavenging of the 2,2-diphenyl-1-picrylhydrazyl (DPPH) radical, hydroxyl radical scavenging (OH), and cupric-reducing ability (CUPRAC) assays. To correlate the obtained results using different assays, an antioxidant index (AOX) was calculated. Furthermore, the antibacterial properties of the obtained nanoparticles were tested by determining the minimum inhibitor concentration (MIC) against two model bacterial species: *Escherichia coli* (Gram-negative bacterium) and *Staphylococcus aureus* (Gram-positive bacterium). Many pathogenic bacteria that lead to disease processes and reside in the environment occur in the form of biofilms. The structure of the biofilm determines the bacteria that form the appearance of additional and unique characteristics, leading to increased resistance to antibiotics and bactericidal compounds. Therefore, a very important part of this study was to determine the antibiofilm potential of the obtained selenium nanoparticles against *E. coli* and *S. aureus* biofilms. The antioxidant capacity of SeNPs and their antibacterial properties have been correlated with their physical properties, such as size, shape, and homogeneity. Attention was placed on the importance of the individual physical parameters of SeNPs for their antioxidant and antibacterial abilities.

## 2. Results

### 2.1. Polyphenolic Composition and Antioxidant Properties of Herbal Infusions

Polyphenolic compounds play a key role in the green synthesis of SeNPs. They are responsible for reducing selenium to its nano form; on the other hand, they stabilize the emerging SeNPs. This is why it is so important to determine the content of polyphenolic compounds in the extracts used for synthesis. The results of the relevant chromatographic analysis are presented in Table 1.

The high content of polyphenolic compounds in herbal extracts determines their high antioxidant capacity. From the point of view of the potential biomedical application of SeNPs, their antioxidant abilities are as important as their dimensions. Often, these properties are significantly higher than the plant extracts used for their synthesis. This was also observed in our research, mainly in the ability to neutralize hydroxyl radicals. Table 2 presents, in detail, the antioxidant capacity determined both for the extracts themselves and the SeNPs obtained with their use. Marking 1/1 1/2 or 1/3 next to the extract refers to the ratio in which it was mixed with water, so its concentration is analogous to that in the synthesis of SeNPs.

The progress of nanoparticle synthesis was monitored by recording UV-Vis spectra. As demonstrated earlier, the location of the absorption maxima of SeNP suspensions can be linked with their dimensions [16]. The corresponding spectra recorded after 60 min of synthesis are presented in Appendix A. The morphology, shape, and size of synthesized selenium nanoparticles were also evaluated via SEM and TEM analyses (Figure 1 and Appendix A). The determined average dimensions and the polydispersity index (PDI), which was determined by the DLS method, are presented in Appendix A.

### 2.2. MIC Analyses

Two bacterial species were selected for the MIC testing, namely, reference strains *Escherichia coli* MC1061 and *Staphylococcus aureus* ATCC 29213. *Escherichia coli* is a Gram-negative, relatively aerobic bacterium that is part of the physiological intestinal microbiota of humans and animals, while *Staphylococcus aureus* is Gram-positive, a relatively aerobic bacterium found primarily on the skin and in the upper respiratory tract of humans. Both bacteria are ranked among the main pathogens responsible for nosocomial infections [17]. The initial plant extracts used to synthesize selenium nanoparticles showed no or weak bactericidal activity against both bacterial species tested. No bactericidal effects were observed for nanoparticle-free extracts of hops and lemon balm regardless of their starting dilution. Enrichment of both extracts with SeNPs produces an antibacterial effect that is slightly stronger against the Gram-positive bacterium *S. aureus*. For both hops and lemon balm, no bactericidal effect was observed against *E. coli* for variant 1/3. The other starting extracts of blackberry, raspberry, and sage show little antibacterial activity. Their enrichment with SeNPs enhances the observed antibacterial effect. They show a slightly stronger effect against *S. aureus*. The antibacterial effect of SeNPs obtained by green synthesis based on blackberry, hops, raspberry lemon balm, and sage extracts was confirmed experimentally. For comparison, the activity of selenium nanoparticles obtained by traditional chemical methods was determined. SeNP AA showed no bactericidal effect against *E. coli* and inhibited the growth of only *S. aureus*. SeNP GA showed a bactericidal effect against both bacterial species tested, which was much stronger against *S. aureus*. The addition of a stabilizer in the form of polyvinyl alcohol did not affect the activity of the tested nanoparticles. Many bacteria, especially pathogens, naturally form biofilm structures. For pathogenic bacteria, this is very important due to their increased resistance to the chemotherapeutics used to control them. This study further analyzed the antibacterial efficacy of the extracted SeNPs against biofilms formed by *E. coli* and the Gram-positive bacteria *S. aureus*. The results of the corresponding MIC analysis are presented in Table 3.

### 2.3. Antibiofilm Effect of Selenium Nanoparticles

Biofilm is a structure formed by bacterial communities, characterized by a complex structure. It is surrounded by an autonomously produced extracellular polymeric substance, EPS (EPS—extracellular polymeric substance), which includes exopolysaccharides, metabolites, nutrients, extracellular DNA, and proteins. Biofilms adhere to biotic and abiotic surfaces [18]. Bacterial cells that form biofilm structures are better protected, less prone to mutations, show stronger resistance to antibiotics and chemotherapeutics, and manifest lower metabolic activity. Thus, the formation of biofilm structures by pathogens is related to the chronic nature of the infection, and bacteria have learned to bypass the body’s defense systems [19]. The most common serious infections caused by biofilms formed by *S. aureus* are linked to the susceptibility to the colonization of implanted medical devices and implants, such as urological and intravascular catheters, heart and joint prostheses, and pacemakers [20]. Risk factors contributing to the development of infection include the nasal carriage of *S. aureus* and damage to the surface of the skin, especially if it involves large areas of skin, such as burns or surgical wounds [21]. Effective ways to combat the resulting biofilm structures and substances that can be used for preventive purposes are still being sought. All selected plants, whose extracts were used for the green synthesis of SeNPs, are characterized by antibacterial, antioxidant, anti-inflammatory, and protective properties, which confirm their wide application in medicine and cosmetology, especially in wound healing and as dietary supplements. The use of compounds of natural origin, rich in bioactive components with reducing and stabilizing properties, for the synthesis of selenium nanoparticles, was aimed at ensuring high reaction yields and obtaining a product with the highest possible bactericidal activity and stability. There is no available literature data on the use of extracts of lemon balm, hops, raspberry proper, clary sage, and folded blackberry for the synthesis of selenium nanoparticles. The research conducted is, therefore, highly relevant. The results of the biofilm analysis are presented in Appendix A. SeNPs obtained with blackberry extract against biofilms formed by *E. coli* show a weak antibacterial effect. No effect of temperature on the quality and activity of the obtained SeNPs was noted. The strongest antibiofilm effect for all concentrations of blackberry extracts (extracts mixed with water at 1/1, 1/2, and 1/3 ratios) was observed for undiluted extracts—100%. At dilutions from 50% to 3125%, the observed effect was somewhat weaker; only the 1/3 ratio gave an inhibition equal to or greater than 40% for all dilutions of SeNPs tested. Pure blackberry extracts inhibit *S. aureus* biofilm growth; the strongest effect was produced by extract mixed at a 1/2 ratio with water, with inhibition at about 40%. Selenium nanoparticles obtained with blackberry extract, regardless of the dilution, showed strong antibiofilm activity. The biofilm formed by *S. aureus* for all tested solutions, regardless of the method of obtaining them and the dilution used, was inhibited very strongly; the inhibition reached about 60% for all tested variants, compared to the biofilm formed by the control strain of *S. aureus*. The SeNPs obtained with blackberry extract, thus, showed a strong antibiofilm effect.

The initial hop extracts showed a weak inhibitory effect on biofilm formed by *E. coli* and *S. aureus*. A slightly stronger effect against *E. coli* was produced by 1/1 and 1/3 diluted extracts. SeNPs obtained with a 1/1 ratio of selenium to hop extract demonstrated the strongest antibiofilm effect against *E. coli*. Dilutions of 100–25% had the strongest effect, with inhibition even reaching above 60%. SeNPs obtained with hop extracts in ratios of 1/2 and 1/3 showed a weak antibiofilm effect against *E. coli*; only undiluted solutions were effective. An interesting effect was observed for the initial hop extracts. Pure hop extracts, regardless of their dilution, did not inhibit the development of biofilm by *S. aureus*; on the contrary, in their presence, biofilm was more readily formed by *S. aureus*. It seems that pure hop extract—devoid of selenium nanoparticles—induces biofilm structure formation by *S. aureus*. Selenium nanoparticles obtained with hop extract in all experimental variants showed a very strong inhibitory effect on biofilm formation by *S. aureus*, which reached even more than 50%. This effect did not depend on the type of extract, the method of obtaining SeNPs, and their concentration. SeNPs extracted with hop extract showed an antibiofilm effect much stronger against *S. aureus*.

The initial pure lemon balm extracts used to synthesize SeNPs showed weak or no antibiofilm effect against *E. coli*. The SeNPs obtained with them showed a much stronger antibacterial effect via the inhibition of biofilm formation by *E. coli*; in particular, when used at a concentration of 100%, the biofilm structure inhibition in all variants reached 60% and the effect did not depend on temperature. The strongest bactericidal effect was shown by SeNPs obtained with the participation of lemon balm extract at a ratio of 1/2 to selenium; a strong antibiofilm effect was shown by all concentrations of SeNPs tested in this variant. Pure lemon balm extracts in all variants tested showed an equally strong inhibitory effect on the development of biofilm formed by *S. aureus*; the inhibition of the structure reached more than 40%, even up to 60%. The SeNPs obtained with them significantly exacerbated this effect. Biofilm formed by *S. aureus* was strongly inhibited in the presence of all tested SeNPs obtained with lemon balm extracts, regardless of the dilution of the extract used and the method of obtaining the nanoparticles. Slightly stronger effects were shown by SeNPs obtained with a 1/1 ratio at a concentration of 100%; *S. aureus* biofilm was inhibited by 80%. Similar effects confirming the strong antibiofilm activity of SeNPs against *S. aureus* were also demonstrated by selenium nanoparticles obtained with 1/2 and 1/3 ratio Se/extracts. All analyzed starting raspberry extracts did not inhibit biofilm formation by *E. coli*. On the contrary, biofilms were positively stimulated to grow. Selenium nanoparticles obtained with raspberry extracts for all variants, regardless of the temperature used, inhibited biofilm formation by *E. coli* at a similar level. For concentrations of 100–25%, the biomass of *E. coli* biofilms was inhibited at a similar level. The crude raspberry extracts mixed with water at ratios of 1/1 and ½ showed a weak antibiofilm effect against *S. aureus*. A much stronger inhibitory effect was produced by the 1/3 ratio solution. Extracts enriched with SeNPs showed an increased antibiofilm effect against *S. aureus* cells, as evident by a reduction in the biomass of their biofilms. The effect of SeNPs was similar for all variants, with a slightly stronger effect shown by SeNPs obtained with a 1/3 Se to extract ratio.

All the initial sage extracts analyzed showed a weak antibiofilm effect against *E. coli.* SeNPs obtained with them showed an increased anti-biocidal effect, evident in the form of the reduction of *E. coli* biofilm mass, only at a concentration of 100%, regardless of the extract and the method of obtaining nanoparticles. Crude sage extracts showed a weak antibiofilm effect against *S. aureus*. At the lowest ratios, 1/1 and 1/2 sage extracts even stimulated biofilm formation by *S. aureus*. SeNPs obtained with them showed an inhibitory effect on biofilm formation by *S. aureus*. It was most strongly observed for SeNPs obtained with sage extract in a 1/3 Se/extract ratio, and for the highest concentrations, biofilm was inhibited by 60%; this effect did not depend on the method of obtaining selenium nanoparticles.

Chemically synthesized selenium nanoparticles (using ascorbic acid) did not inhibit the biofilm formed by *E. coli*. Against *S. aureus*, the antibiofilm effect was marked strongly for concentrations of 100–12.5%. Selenium nanoparticles obtained with gallic acid at 100% and 50% concentrations slightly inhibited the biofilm formed by *E. coli*. Biofilm formed by *S. aureus* in the presence of SeNPs GA was almost completely inhibited for all concentrations analyzed.

## 3. Discussion

### 3.1. Characterization of Herbal Extracts

Although the mechanism of green SeNP synthesis is complicated, secondary plant metabolites, including polyphenolic compounds, seem to play a key role in this process. It is postulated that the first stage of synthesis is the saturation of metal cations derived from the salt, resulting in the formation of hydroxyl complexes [22]. The second stage is a crystallite growth of metal with oxygen species starting to originate. This process continued until the activation of the capping agent from the plant extract, which eventually stopped the growth of high-energy atomic growth planes. Reducing agents from plant extracts are electron donors to metal ions, converting them into nanoparticles. Thus, polyphenolic compounds are not only important at the stage of the synthesis itself, but also after it, because they can prevent the aggregation of nanoparticles and, thus, promote the production of smaller particles. From this point of view, the use of a plant extract rich in polyphenolic compounds (and, thus, high antioxidant capacity) is the first step to the effective green synthesis of SeNPs. The polyphenolic composition is presented in detail in Table 1, while the proper antioxidant activities of an extract are shown in Table 2. All extracts used for the study were characterized by a high content of flavonoids and polyphenolic acids. Understandably, the individual extracts differed in both the content of individual compounds and the polyphenol profile itself. One undeniable difficulty in the green synthesis of nanoparticles is the difference in the content of individual polyphenolic compounds, even in the same species but from different crops. The differences in chemical composition may be due to a variety of reasons, ranging from climate and geography to the difference in the specificity of the extraction procedures [9]. This significantly hinders the process of unification and optimization as well as the repeatability of the green NP syntheses being developed, making them more difficult to control than classical chemical methods of synthesis. Borros et al. [23] found only one flavonoid, luteolin-3-O-glucuronide, in cultivated, in vitro cultured, and commercial samples of *Melissa officinalis*. In our study, many more flavonoids were detected, which confirms our previous study [24] and highlights the problem that was mentioned earlier. Some hydroxycinnamic acids (caffeic acid, chlorogenic acid, and p-coumaric acid) and hydroxybenzoic acid derivatives (gallic acid and *p*-HBA) were found in the studied extracts. Chlorogenic acid was found in the infusions at the highest level, but a large dispersion between its concentrations in different extracts was observed. p-Hydroxybenzoic acid (pHBA) was also present at significant levels. In plants, caffeic acid is formed from p-coumaric acid and then transformed into ferulic acid [25]. All of them were detected in the extracts, except lemon balm infusion, where the concentration of ferulic acid was below the limit of detection. The extracts used were also tested for their antioxidant compatibilities. For better comparison, the antioxidant potential was calculated as a percentage of the average antioxidant capacity of a given sample compared to the highest one. Then, the AOX index was obtained for each sample as the sum of the individual indexes divided by the number of tests (in the case of this study 4). All the analyzed extracts were characterized by high antioxidant capacity, to the extent that comparing their AOX turned out to be impossible due to their similar values. For the ratio of extract to water, 1/1 for all studied samples, the index was in the range of 97–98%. The exception was sage extract; this value was 98.7%. Differences between extracts can be seen only when comparing the results of individual tests. The Folin–Ciocalteu method allows one to determine the total content of polyphenolic compounds in a given sample. The results obtained with this method increase in series: hop < raspberry < sage < blackberry < lemon balm. The reducing properties of the extracts determined using the CUPRAC method were in the range of 1.20–1.50 mmolTr/L. The results demonstrating the ability to neutralize free radicals allow for the differentiation of individual extracts. The results obtained for the method using the DPPH radical prove that the hop extract has the lowest antioxidant properties. This is in line with the results obtained from the Folin–Ciocalteu method, based on which, it can be concluded that hop extract contains the lowest total polyphenol content. The remaining extracts were characterized by similar abilities to neutralize DPPH radicals (the values obtained for the extract mixed with water at a ratio of 1 to 1 were in the range of 0.850–0.875 mmolTr/L). It should be emphasized that the DPPH method is the most commonly used method to determine the antioxidant capacity of samples; however, the radical itself is a model radical that does not occur in living organisms. Therefore, from the point of view of potential biomedical applications of the tested extracts, as well as the nanoparticles themselves, their ability to neutralize hydroxyl radicals is much more reliable. In this case, the analyzed extracts are divided into two groups. Those whose ability to neutralize hydroxyl radicals are relatively low and amount to less than 30%, and those for which this value is above 85%. The lowest ability to neutralize hydroxyl radicals was shown by raspberry extract (26.2 ± 1.10%) and sage extract (30.4 ± 1.03%). The hop extract was characterized by medium capacity (55.3 ± 2.07%), despite the lowest content of polyphenolic compounds. However, it should be mentioned that the results obtained for all spectrophotometric methods used to determine the antioxidant capacity are the results that are affected by all chemical compounds present in the sample, including those that are not polyphenol compounds. Blackberry and lemon balm extracts showed the highest ability to neutralize hydroxyl radicals (88.0 ± 3.21% and 96.1 ± 3.47%, respectively).

### 3.2. Synthesis and Characterization of Obtained SeNPs

The tested extracts were used to synthesize selenium nanoparticles (SeNPs). The UV-Vis spectra recorded after 60 min of mixing the reagents show a clear absorption band at the wavelength in the range of 250–300 nm (Appendix A). A slight shift in the absorption maximum toward higher wavelengths was observed with an increasing concentration of the plant extract in the reaction mixture. This phenomenon is observed for all syntheses, regardless of the extract used, and may suggest a change in the size of the SeNPs formed along with a change in the ratio of reagents used for the reaction. Lin and Chris-Wang related the location of the absorption maxima of selenium nanoparticles to their sizes [16]. The authors proved that 18.1 ± 6.7 nm-sized SeNPs exhibit an absorption maximum of around 250 nm. With the increase in the size of nanoparticles, the absorption maximum is shifted, and for SeNPs with a size of 100 nm, it is located at around 350 nm. The spectra recorded by us suggest the formation of SeNPs with dimensions smaller than 100 nm, in the case of a 1:1 ratio of reactants. These initial observations were confirmed by DLS analysis. The results obtained from dynamic light scattering confirmed the formation of nanoparticles with dimensions of about 100 nm for a ratio of reagents of 1:1. The size of the obtained particles increased with the increase in the concentration of the extract used for the synthesis. The nanoparticles obtained using the method in which the reagent ratio was 1/2 had dimensions of about 200 nm, according to DLS. Further increasing the concentration of reagents did not result in any significant changes in the size of the obtained SeNPs. This is in agreement with the recorded UV-Vis spectra. Similar results were obtained for samples subjected to additional heating after synthesis, which was supposed to ensure a more spherical shape of the nanoparticles [26]. However, literature reports on heating are contradictory; for example, Zhang reported that heating treatment (1 h at 90 °C) caused aggregation of SeNPs into larger sizes and rods, which led to a significant reduction of their bioactivity in mice [27]. Nevertheless, a negative correlation was found between the ratio of reagents used for the synthesis and the size of the SeNPs formed. Correlation coefficients for this relationship were high for all extracts used, and the values of their correlation coefficients were higher than 0.9. The exception was nanoparticles obtained with the use of hop extract, for which the value of the coefficient was 0.353, but the shift in the absorption maximum, in this case, is minimal (Appendix A). This trend was observed for all herbal extracts used. The size of the selenium nanoparticles is a very important parameter if they are going to be used in medicine because the availability of selenium from nanoparticles also depends on their size. Se from the 36 nm particles with a higher surface area per dosed mass unit was more bioavailable than from the larger particles with a diameter of 90 nm [28]. In general, nanoparticles larger than 100 nm cannot freely cross the barrier of the cell membrane; hence, their size is so important. However, it should be noted that the obtained nanoparticles are so strongly stabilized by other compounds contained in the post-reaction mixture that it is impossible to clean them thoroughly. On the other hand, treating the post-reaction mixture as a suspension of SeNPs, we should be aware that the nanoparticles form clusters and are covered with other substances, which may cause misinterpretation of the results. For this reason, a much more sensitive technique than DLS should be used to accurately determine their size. When the image of simple TEM is studied, it can be concluded that nanoparticles with dimensions lower than 100 nm are obtained when the ratio of reagents used for the synthesis is 1:1. The exact data are presented in Appendix A. The results obtained from the TEM confirm those of the previously used methods. Only nanoparticles synthesized with a 1:1 ratio of reagents were smaller than 100 nm and only they could potentially be used for biomedical purposes. Increasing the concentration of individual extracts also does not bring the desired effect, but rather increases the size of the nanoparticles. Also, the heating of the post-reaction mixture increased the size of the nanoparticles. This effect was observed for each extract and each reagent concentration ratio used. According to the literature, the heating of the post-reaction mixture is supposed to increase the sphericity of the obtained SeNPs. In turn, sphericity increases the antioxidant capacity of nanoparticles. No such effect was observed in our study. Moreover, heating led to the aggregation of nanoparticles and increased their size. It also harmed their shape. As a result of heating, they changed from a spherical shape to nanorods or large clusters with a more angular shape. The described situation is illustrated in Appendix A for the hop and sage example. Many authors reported that they have obtained spherical SeNPs, but other shapes, such as elongated cylinders, polygons, and granules were also observed [29,30,31].

The course of green SeNP synthesis is more difficult to control than in the case of chemical synthesis. This is due to the very idea of green synthesis, which uses a plant extract, i.e., a mixture of many plant-derived compounds, which, in different ways, affect the course of the reaction and, thus, the properties of the obtained NPs. This also translates into the homogeneity of the obtained nanoparticles. It is also an important parameter affecting the antioxidant capacity of SeNPs, which we reported earlier [4]. To assess the homogeneity of the obtained nanoparticles, the polydispersity index, PDI, was determined using the DLS method. According to the theory, the sample can be considered monodisperse when the PDI value is less than 0.1. The PDI values for all SeNPs obtained are summarized in Appendix A. The lowest PDI values were obtained for SeNPs synthesized at a 1:1 ratio. However, values close to 0.1 were determined only for hops and lemon balm. For the remaining herbs, they were higher but within the range of 0.150. Increasing the concentration of the extract used for the reaction caused a decrease in the homogeneity of the synthesized SeNPs and, thus, an increase in the PDI coefficients. The heating of the post-processing mixture also harmed the homogeneity of the SeNPs. For each reagent ratio and extract, the PDI values were higher when the nanoparticles were subjected to additional heating. The smallest homogeneity and, thus, the highest PDI coefficient were characterized by nanoparticles obtained using sage extract at a reagent ratio of 1:3 and subjected to additional heating. The corresponding PDI value, in this case, was 0.392. Generally, only laser ablation and microwave irradiation, which are physical methods of obtaining SeNPs, allow for the production of nanoparticles with narrow size distribution. Chemical or biological synthesis methods produce a wide range of particle sizes [32].

### 3.3. Antioxidant Properties of Obtained SeNPs

All obtained nanoparticles were tested for their antioxidant properties and the collected results are presented in Table 2. The lower antioxidant capacity determined by the Folin–Ciocalteu method, compared to the corresponding extracts, suggests that polyphenolic compounds play a significant role in SeNP synthesis. These compounds were consumed in the synthesis reaction, hence their lower content in the post-reaction mixture. A similar situation can be observed when analyzing the results obtained with the CUPRAC method. In terms of biomedical applications, the ability of SeNPs to neutralize free radicals is crucial, as reactive oxygen species (ROS) in living organisms contribute to oxidative stress. Increased oxidative stress is associated with metabolic risk factors and may contribute to the development of several diseases. Our research confirms that SeNPs are great nanoantioxidants. In the case of the results obtained using the DPPH radical method, SeNPs did not always show higher antioxidant capacity than the extract used for the synthesis. Such a situation was observed, for example, for lemon balm or blackberry. The results also lack a trend between the ability to scavenge DPPH radicals and the heating of the reaction mixture. In some cases, heating increased the antioxidant capacity of SeNPs (e.g., hops), in others, it depended on the ratio in which the reagents were mixed (e.g., lemon balm). Completely different conclusions are provided by the results in determining the ability to neutralize hydroxyl radicals. It should be recalled that these are more important from the point of view of potential SeNP applications, as OH radicals occur in living organisms. In many cases, the ability to neutralize hydroxyl radicals by SeNPs was higher than that of the extract used for their synthesis. In the case of raspberry and sage, the nanoparticles had three times higher antioxidant capacity than the corresponding extracts. Heating the mixture had a different effect on the demonstrated ability to neutralize OH radicals. This influence also depended on the ratio of reagents used for the synthesis. The relationship between the ratio of reagents and the ability to neutralize OH radicals was clear for the extracts of blackberry and hop, and the obtained correlation coefficients were high (0.911 and 0.989, respectively). However, in the case of raspberry and sage, there was a negative correlation and here: with the increase in the concentration of the extract in the reaction mixture, the ability of the obtained nanoparticles to neutralize hydroxyl radicals decreased. The determined correlation coefficients were, respectively, −0.985 (raspberry) and −0.965 (sage). A moderate correlation coefficient was obtained for lemon balm, and it was 0.631. An attempt was also made to correlate the AOX coefficient with the size of the nanoparticles. However, in this case, moderate correlation coefficients were obtained, indicating a negative correlation. This trend was the same for heated and unheated samples. The exception was sage, for which the determined correlation coefficients between the AOX index and the value of SeNPs amounted to 0.990 for unheated samples and 0.973 for heated samples. An analogous correlation determined for SeNPs obtained from the raspberry extract and not subjected to heating was characterized by a high correlation coefficient of 0.975.

### 3.4. Antibacterial Properties of Obtained SeNPs

The resistance of microorganisms to antibiotics is currently a very important medical problem. Nanostructured inorganic compounds are increasingly emerging as a potential tool in the fight against antibiotic-resistant bacteria. In addition to metals, metal oxides, and non-metals, various types of nanoparticles have been found, such as AgNPs, AuNPs, and SeNPs. Selenium nanoparticles are classified as agents with significant antibacterial activity, especially for chronic and nosocomial infections caused by pathogenic bacteria. To obtain SeNPs using the green synthesis method, widely available plants with recognized therapeutic potential were selected. The study showed that pure blackberry, raspberry, and sage showed weak bactericidal effects against *E. coli* cells. They had a slightly stronger effect against *S. aureus* cells, and a stronger effect was produced by pure blackberry and sage extracts. Pure extracts of hops and lemon balm did not inhibit the growth of free-living *E. coli* and *S. aureus* cells. The literature data suggest that the aqueous extract of lemon balm has antibacterial potential, as it is effective against free-living *E. coli* and *S. aureus* cells [33]. It should be remembered that the composition of the extract may vary, as it depends on the plant itself, its geographical location, age, method of extract extraction, or harvesting period. All the mentioned factors affect the content of compounds responsible for the plant’s antibacterial properties [34]. In contrast, a study by Kramer et al. showed that Gram-positive bacteria are more sensitive to aqueous extracts of hops [35]. It has also been shown that the method of extraction of active substances from hops is of extraordinary importance in studying their antibacterial properties, as different MIC activities are shown by essential oil, infusion, and decoction of hops [36]. This means that there are substances in the plant that show biocidal properties, but their effectiveness depends on the method of their isolation. The SeNPs synthesized in the study with blackberry, hop, and sage extracts showed the strongest bactericidal activity against the bacteria tested. The presence of SeNPs in the post-reaction solution resulted in an increased bactericidal effect against *E. coli* and a much stronger effect against *S. aureus*. The antibacterial effect of SeNPs obtained by green synthesis was confirmed experimentally.

Serov et al. noted that the value of the minimum inhibitory concentration in antibacterial studies strictly depends on the method of SeNP synthesis [32]. The minimum inhibited concentration for effective antibacterial action did not exceed 100 µg/mL for SeNPs obtained via physical synthesis. On the other hand, when using microwave generation of nanoparticles, the MIC is approximately 100–300 µg/mL. The authors highlight that, in the near future, we should expect clarification of the data presented.

Analyses conducted to analyze the structure of biofilms showed significantly weaker antibiofilm activity of crude plant extracts. A weak antibiofilm effect was shown by crude lemon balm extract against *S. aureus* cells, while hop extract, on the contrary, induced biofilm formation by this bacterium. All plant extracts enriched with selenium nanoparticles showed a very strong antibiofilm effect against Gram-positive *S. aureus* cells. The strongest effect was observed for extracts of lemon balm and hops enriched with SeNPs, and a slightly weaker effect was observed for extracts of blackberry, sage, and raspberry enriched with SeNPs. The strong inhibitory effect on biofilm formation by *S. aureus* did not depend on the 1/1, 1/2, or 1/3 reagent ratios used, as well as the concentration and temperature used. For all experimental variants tested, biofilm formation by *S. aureus* was inhibited equally strongly. For lemon balm post-reaction mixtures, the inhibition reached 80% relative to the control. Analysis of the biofilm formed by *E. coli* showed that the crude plant extracts have no—or little—antibiofilm activity, observed mainly at high concentrations. A slight inhibitory effect on the biofilm formed by *E. coli* was observed for hop and sage extracts at the 1/1 reagent ratio. The post-reaction mixture more strongly inhibited the biofilm formed by *E. coli*, but this effect was weaker than that observed for *S. aureus*. Mixtures containing SeNPs and residues of lemon balm, raspberry, and hop extracts had the strongest effect, slightly weaker than sage and blackberry. Selenium nanoparticles against *E. coli* cells acted more strongly in extracts with ratios of 1/2 and 1/3; the strongest inhibitory effect was observed for concentrations of 100–25%, but the effect did not depend on the temperature used. As expected, SeNPs obtained with the use of herbal extract at ratios of 1/2 and 1/3, had a positive effect on the activity of the analyzed mixtures, probably due to the higher concentration of reductants and stabilizers derived from the extracts, as they are responsible for the quality of the synthesized nanoparticles. The stronger activity of the suspensions against *S. aureus* biofilms, compared to biofilms formed by *E. coli*, is due to the difference in the structure and permeability of the cell membranes of Gram-positive and Gram-negative bacteria, directly affecting their susceptibility to SeNPs and plant-derived compounds. Analyses demonstrating the antibiofilm effect of plant extracts are available in the literature. Alcoholic extracts and essential oils have been analyzed; Gómez-Sequeda et al. proved that the essential oil extracted from sage is capable of inhibiting biofilm formation by *E. coli* by up to 30%, and *S. aureus* by up to 36% [37]. However, it should be noted that the concentration of components with bactericidal activity in oils is much higher than in aqueous extracts, so the oil itself can be used as an effective antibacterial agent at lower concentrations. Strugala et al., on the other hand, showed that both aqueous and methanolic blackberry extracts exhibit strong biocidal activity against *E. coli* [38]. In addition, they hinder bacterial adhesion to urinary tract epithelial cells, due to which, the authors recommend the use of blackberry extracts as dietary supplements aimed at reducing the risk of urinary tract infections. In the study, the effect of blackberry extract on the biofilm formed by *E. coli* was found to be moderate, and its antibiofilm activity increased with the sample incubation time; the increased sensitivity to the test substances may be due to the depletion of nutrient compounds in the ongoing cultures [38]. To compare the efficacy of the acquired nanoparticles, we decided to study the antibiofilm activity of SeNPs acquired by classical chemical methods. All tested SeNPs showed a strong inhibitory effect on the biofilm formed by *S. aureus* regardless of the use of a stabilizer in the form of polyvinyl alcohol. SeNP AA did not inhibit biofilm formed by *E. coli*, SeNP GA slightly inhibited biofilm formed by *E. coli* only at high concentrations.

The obtained results confirmed the strong antibacterial effect of the obtained SeNPs. Research on the mechanisms of action in vivo on bacterial cells is ongoing. However, it seems that in the case of SeNPs obtained in this study, the key mechanisms of action will be related to the generation of oxidative stress in cells and the uncontrolled synthesis of reactive oxygen species. It is known that antimicrobial nanoparticles can damage bacterial cells through multiple pathways. This multimodal antimicrobial mechanism of action makes NPs very attractive, as bacteria are expected to have difficulty developing resistance to multiple forms of attack [39]. Selenium nanoparticles can cause the degradation of proteins due to their bactericidal action [40] and inhibit the activity of the dehydrogenase enzyme [41]. Their destructive effect on the bacterial cell membrane has been widely described in the literature. SeNPs contribute to the inactivation of the natural mechanisms of the membrane transport of ions and nutrients through cell walls, blocking the vital activity of the cell [42]. Moreover, the slow emission of selenium ions from the surface of nanoparticles can lead to their interaction with the -SH, -NH, or -COOH functional groups of proteins and enzymes and the subsequent loss of their tertiary and quaternary structures and functions [41]. All these issues will be determined for the SeNPs obtained by us in further stages of research.

### 3.5. Antibacterial and Antioxidant Activities of SeNPs versus Their Physical Parameters

The obtained research results allow one to conclude that there is no clear correlation between the antioxidant and antibacterial abilities of the obtained selenium nanoparticles. When blackberry extract was used for the synthesis, the maximum antibacterial abilities were determined for SeNPs obtained for the reaction carried out at a substrate ratio of 1/2. At the same time, these nanoparticles showed the lowest antioxidant capacity. The maximum of demonstrated properties cannot be related to their size or homogeneity. In the case of nanoparticles obtained using hop extract, high antioxidant properties were also determined for samples for which no inhibition of biofilm formation was observed. Similar contradictions are observed for nanoparticles obtained using sage, raspberry, and lemon balm. Some authors observed that as the size of nanoparticles decreases, on average, they become more effective against viruses [32]. However, they studied SeNPs obtained via physical synthesis, so the sample matrix was much easier to control than in the case of our study. These observations emphasize that in the case of green synthesis, in which one of the reactants—the extract—is a mixture of biologically active compounds, it is difficult or even impossible to predict the properties of the obtained product. The observed effect is the result of the properties of the nanoparticles themselves and the sample matrix, i.e., the remains of the plant extract after synthesis. The effect of selenium dioxide on the biologically active components of the sample matrix cannot be ignored. Shangpliang et al. demonstrated that the presence of these reactive forms of selenium leads to the formation of various potentially active compounds such as α-oxo selenoamides [43]. It cannot be ruled out that such a process also occurs in the case of the described methods of green SeNP synthesis. Therefore, this may be a potential reason for the lack of correlation between the antioxidant and antibacterial properties of SeNPs and their size, shape, and homogeneity. Their properties can be influenced not only by the sample matrix-derived substances adsorbed or even chemically bound to the nanoparticle structure, but also by the product of their reaction with selenium dioxide. This aspect requires further research. Therefore, there is no single universal method of green synthesis using herbal extracts that would achieve maximum antioxidant and antibacterial properties at the same time. The synthesis method used should, therefore, be adapted to a specific purpose.

## 4. Materials and Methods

### 4.1. Reagents

Sodium selenite (Na_2_SeO_3_) used in the synthesis of selenium nanoparticles was purchased from Merck-Sigma (Steinheim, Germany). Standards of polyphenolic compounds used in the analysis of herbal infusions (qualitative and quantitative analysis), as well as all reagents used to determine the antioxidant capacity, were also purchased from Merck-Sigma. Methanol (MeOH) and acetonitrile (ACN), used as the organic components of the mobile phase in HPLC analysis, were also by Sigma-Merck. Water used at every stage of the experiment was obtained from the Milli-Q system (Millipore, Bedford, MA, USA).

### 4.2. Herbal Samples

The dried herbs used in the experiment came from one manufacturer: Kawon (Gostyń, Poland). Before use, each was ground in a ball mill. Then, 5 g of dried material was weighed out and placed in a glass beaker. Then, 50 mL of boiling water was added, and the beaker was covered with a glass cover. The brewing was carried out for 30 min using a magnetic stirrer (the intensity of mixing was 200 rpm). Before further use, all of the extracts were filtered through a filter paper and used for the SeNP synthesis or were filtered through a 0.22 µm filter and analyzed by HPLC.

### 4.3. Chromatographic Analysis of Polyphenolic Compounds in Studied Infusions

Chromatographic separation of polyphenol was carried out on ZIC-HILIC (100 × 2.1, 3 μm) from Merck. The mobile phase consisted of ACN and water, delivered at 0.2 mL/min in gradient mode, as follows: 0–4 min 98% B, 6–7 min 90% B, 8–84 min 80% B, 8.4–12 min 50% B, and 13–20 min 98% B. Compounds were identified based on the comparison of their retention times and *m*/*z* values obtained by MS and MS2 with the mass spectra for the standards.

### 4.4. Green and Chemical Syntheses and Characterization of Obtained SeNPs

Green synthesis of SeNPs was based on the reduction of Na_2_SeO_3_ with the herbal extracts. The sodium selenite solution (0.1 mol L^−1^) was prepared before the synthesis. The methodology was as follows: 15 mL of deionized water was added to 2.5 mL of sodium selenite solution and placed at the magnetic stirrer. The intensity of mixing was 1000 rpm. After that, 2.5, 5, or 7.5 mL of the extract was added dropwise. Such a procedure enables carrying out the reaction at different ratios of the extract concentration to the constant concentration of selenium (Se/infusion ratio 1:1, 1:2, and 1:3, respectively). The reaction mixture was stirred for 1 h at room temperature (25 °C). Parallel synthesis was performed for each ratio of reagents and the post-reaction mixture was heated in a 70 °C water bath for 1 h. This procedure was aimed at increasing the sphericity of the SeNPs obtained. Due to the strong stabilization of the SeNPs by the compounds present in the extract, problems with their separation occurred. It was decided to conduct further research on the obtained SeNP suspension without their isolation from the post-reaction mixture (herbal infusion).

In parallel, conventional chemical synthesis of selenium nanoparticles was carried out, using ascorbic acid and gallic acid as selenium reducers. All of the standard solutions were prepared before the synthesis. The selection criteria were that these compounds be present in plant material, including herbal extracts, and not be considered toxic. Briefly, 20 mL of Na_2_SeO_3_ solution (5 × 10^−3^ mol L^−1^) was placed in a beaker with a magnetic stirrer. Then, 10 mL of 4 × 10^−2^ mol L^−1^ ascorbic acid (AA) or gallic acid (GA) solution was added dropwise. After one hour of mixing at room temperature (25 °C), 70 mL of Mili-Q water was added.

The UV-Vis spectra in the range of 250–900 nm were recorded for each obtained suspension of SeNPs. For the measurements, each sample was diluted 50 times. To avoid the interferences related to the color of the herbal infusion used for the synthesis, the spectra were recorded using the appropriate infusion as a blank. The concentration was equal to that in the reaction mixture. The measurements were performed using Perkin Elmer spectrophotometer Lambda 20, equipped with cuvettes of 1 cm in length.

The size of the obtained SeNPs was investigated using dynamic light scattering (DLS), performed on Mastersizer 2000 (Malvern, Panalutical, UK) with a wet sample dispersion unit (Hydro 2000 MU, Malvern, Panalutical, UK). The instrumentation allows measuring particles larger than 0.01 µm (10 nm).

The size and shape of SeNPs were studied by transmission electron microscopy (TEM) with a TALOS F200 model (Thermo Fisher Scientific, Waltham, MA, USA) working at an accelerating voltage of 200 kV. A drop of bright red solution containing synthesized selenium nanoparticles was placed on a copper grid and then air-dried before the examination. The obtained results were processed in the iTEM program, which is part of the apparatus software. Scanning electron microscopy (SEM) was also involved, using a field-emission SEM (Merlin Zeiss, Montreal, Canada) for images and the morphology of SeNPs. Before measurements, samples were plasma-sputtered with a few-nanometers-thick Au/Pd layer. The obtained results were processed in the program iTEM, which is part of the apparatus’s software.

### 4.5. Antioxidant Activity Measurements

#### 4.5.1. Hydroxyl Radical Scavenging

To test the ability to neutralize hydroxyl radicals, the method described by Smirnoff and Cumbes [44] was used. The reaction mixture consisted of 1 mL of iron sulfate (1.5 × 10^−3^ mol L^−1^), 0.7 mL of hydrogen peroxide (6 × 10^−3^ mol L^−1^), and 0.3 mL of sodium salicylate (2 × 10^−2^ mol L^−1^), and was mixed with 1 mL of a particular sample (herbal extract or SeNP solution). The sample was incubated at 37 °C for 60 min. After that, the absorbance was measured at 562 nm. The obtained results are expressed as the percentage of OH radical scavenging.

#### 4.5.2. Total Phenolic Content—Folin–Ciocalteu Assay

The Folin–Ciocalteu (FC) assay was also performed to determine the content of polyphenolic compounds in herbal infusions, which are responsible for the reduction in selenium to its nano form. Determining the total phenolic content in the post-reaction mixture can also be a measure of the participation of polyphenolic compounds in the synthesis process. In this method, 1 mL of the sample was mixed with 0.1 mL of FC reagent and 0.9 mL of water. After 5 min, 1 mL of 7% solution of Na_2_CO_3_ and 0.4 mL of water were added. After another 10 min, the absorbance was measured at 765 nm. The results are expressed as the gallic acid (GA) equivalent.

#### 4.5.3. The Reducing Capacity of the Samples—CUPRAC Assay

The capacity to reduce cupric ions was determined using the CUPRAC method described previously by Apak [45]. Briefly, 1 mL of copper chloride (1 × 10^−2^ mol L^−1^) was mixed with 1 mL of neocuproine methanolic solution (7.5 × 10^−3^ mol L^−1^) and 1 mL of ammonium acetate buffer (1 M, pH 7). Then, 0.5 mL of herbal infusion or SeNP suspension and 0.6 mL of water were added. The mixture was incubated at 50 °C in the water bath for 20 min. Absorbance against the blank was measured at 450 nm. The obtained are expressed as a Trolox equivalent (TRE) in μM.

#### 4.5.4. DPPH Radical Scavenging Assay

Scavenging of free radicals was also performed using the DPPH assay. Briefly, 0.1 mL of a sample was added to 2.4 mL of the methanolic radical solution (9 × 10^−5^ mol L^−1^). The decrease in absorbance was measured 30 min after mixing reagents at 518 nm. The results are expressed as a Trolox equivalent (TRE) in μM.

Each sample in each assay was analyzed in triplicate.

#### 4.5.5. Antioxidative Index (AOX)

To correlate the obtained results using different assays, an antioxidant index (AOX) was calculated. For this purpose, the procedure described by Seeram et al. [46] was applied. Firstly, an appropriate index was established for each test that was used in the study. It was calculated as a percentage of the average antioxidant capacities of a given sample compared to the highest (sample score/best score × 100). Then, the AOX index was obtained for each sample as the sum of the individual indexes divided by four (the number of tests).

### 4.6. Antibacterial and Antibiofilm Activity Measurements

#### 4.6.1. Strains of Microorganisms Used in the Work

The bacterial strains used in this study were *Escherichia coli* MC1061 and *Staphylococcus aureus* ATCC 29213 from the collection of the Institute of Microbiology, Department of Biology, UW. LB medium—Lysogeny Brith (BioMaxima, Lublin, Poland) was used for bacterial culture. Solid medium was obtained after solidifying LB with 1.5% agar (BioMaxima, Lublin, Poland).

#### 4.6.2. Analysis of the Sensitivity of *E. coli* and *S. aureus* to SeNPs

The MIC (minimal inhibitory concentration) is the lowest concentration of an antibacterial compound that inhibits the growth of the free-living bacteria of the species tested. The test was performed on titer plates by serial microdilution using the classical method (according to the Rohm and Haas method, RH-Europe RM-001–0608). Material taken from 24 h cultures on LB solid medium was suspended in saline solution to obtain a suspension with a density of 0.5 McFarland. Density was measured using a densitometer. Test extracts were then prepared according to Section 2.3, at a final concentration of 100%. The double dilution test was performed in 96-well titration plates, and 100 µL of the LB medium was added to the wells of the first column. Then, a series of double dilutions was prepared, transferring 100 µL of the suspension to subsequent columns. The last column was controlled and contained the medium without the addition of the tested extracts. The prepared double dilution series were inoculated with 25 µL of *E. coli* or *S. aureus* suspensions, the final concentrations of test extracts were: 100%, 50%, 25%, 12%, 6.25%, and 3.125%.

The plates were left to grow at 37 °C, shaking the culture vigorously, for 24 h. The MIC value, according to CLSI (Clinical and Laboratory Standards Institute) guidelines [47], was considered to be the lowest concentration of the antibacterial compound at which no bacterial growth was observed by the turbidity of the medium. The test was repeated a minimum of three times for each bacterium tested.

#### 4.6.3. Study of the Effect of SeNPs on the Formation of Biofilms by *E. coli* and *S. aureus*

Titer plates for biofilm studies were prepared similarly to MIC analyses. Analyses were carried out according to the method described by [48]. Cultures of *E. coli* and *S. aureus* for biofilm determinations were run statically at 37 °C. The classical crystal violet staining method was used to stain the biofilm mass deposited on the polystyrene surface. Non-adherent cultures located in the wells above the biofilm were gently removed by rinsing in water, and the resulting deposit was dried at 37 °C. The plates were then stained with a 0.1% crystal violet solution at room temperature for 10 min. After the allotted time, the dye was removed by rinsing the plate twice in distilled water. The plate was dried again at 60 °C; 95% ethanol was used to suspend the biofilm. The plates were incubated undercover at room temperature for 15 min, with continuous stirring. The extracted mixtures were transferred to a clean 96-well titration plate. Absorbance was measured at 570 nm in a titer plate reader—TECAN Sunrise (microplate reader).

## 5. Conclusions

Despite the highly developed advancements in medicine, microbial infections remain a significant factor in morbidity and mortality worldwide. The development of nanotechnology presented a possible solution to this problem. Unfortunately, recent data indicate the possibility of bacterial resistance to metal nanoparticles and metal oxides [49]. Thus, a search is underway for non-metallic nanoparticles, characterized by antibacterial and antioxidant capacity. SeNPs are such types of nanoparticles. The study of their antimicrobial properties is a new field of science. More than 95% of research papers on this subject have been published in the last 10 years [32]. That is why it is so important to exchange results in scientific discussions, even when they are difficult to interpret. The research described in this work is preliminary and will certainly be continued, even for the accurate interaction between the received SeNPs and the sample matrix. At this stage, one cannot yet talk about the transfer of described syntheses from the laboratory scale to industrial use, and even more so about their medical use.

In summary, selenium nanoparticles obtained by the green synthesis method with the participation of plant extracts from hops, raspberry, sage, blackberry, and lemon balm, showed a strong antibacterial effect, which is comparable to the effect of chemically obtained SeNPs. The tested SeNPs showed a much stronger bactericidal effect against *S. aureus* cells, as expressed by both lower MIC values and strong inhibition of biofilms formed by *S. aureus*. Against *E. coli* cells, the antibacterial effect is slightly weaker but strong enough to be used therapeutically. In vivo studies conducted on bacterial cells coincide with the preliminary characterization of green synthesized SeNPs, as no beneficial effect of temperature on their activity was observed. For most of the extracts tested, the activity of SeNPs obtained with their participation did not differ significantly for different dilution variants. For most, the strongest antibacterial and antibiofilm activities were observed for the 1/1 ratio. Despite the lowest polyphenol content and the weakest antioxidant properties, SeNPs from hops, raspberry, and sage showed the strongest activity against *E. coli* cells. It seems that against Gram-negative cells, the antibacterial activity of Se nanoparticles may be due to their ability to neutralize hydroxyl free radicals, which is the highest for the mentioned extracts. Against *S. aureus* cells, the most potent activity was demonstrated by extracts of lemon balm and blackberry, which contained higher concentrations of polyphenols, accompanied by strong antioxidant activity. The study confirms the effectiveness of green synthesis in obtaining SeNPs with high antibacterial potential. The probable mechanism of action of the nanoparticles is due to the disruption of mechanisms that enable the removal of free radicals from cells. For infections of unknown etiology, to obtain the greatest bactericidal efficacy, it would be advisable to use mixtures derived from different extracts. However, for their medical applications, cytotoxicity and genotoxicity tests would be necessary, as well as tests on pathogenic bacteria, including anaerobic and microaerophilic microorganisms, which are our immediate plans. However, some authors suggest that biologically obtained SeNPs are not as toxic as those obtained using chemical methods, potentially making them much safer for use as potential therapeutics and preventing the risk of overdose [50].

The obtained SeNPs showed high antioxidant capacity and, thus, can be useful in dealing with oxidative stress. However, there was no correlation between antioxidant and antibacterial properties. Green synthesis of SeNPs is a promising alternative to chemical synthesis, but the main disadvantage of this purpose is the lack of correlation between the physical parameters of the obtained SeNPs and their antioxidant and antibacterial properties. This phenomenon is due to the impact of biologically active compounds present in the sample matrix (in herbal extract). In chemical synthesis, this aspect is easy to predict and control. However, it should be emphasized that the mentioned matrix stabilizes the obtained SeNPs without the need to add chemical stabilizers, which highlights the potential of green synthesis methods.

## Figures and Tables

**Figure 1 molecules-29-01686-f001:**
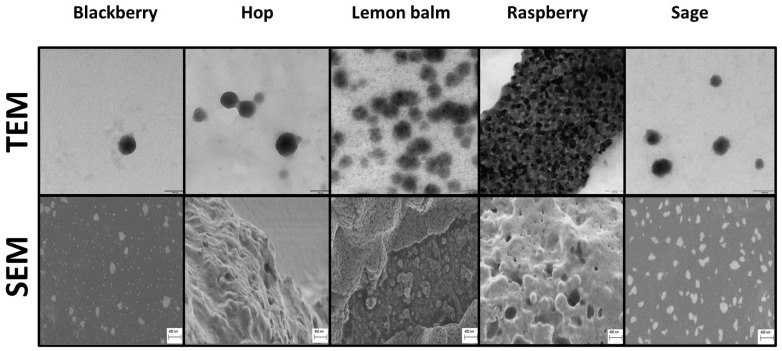
SEM and TEM images of SeNPs obtained by green synthesis, which show the highest ability to neutralize hydroxyl radicals. In all cases, it was a 1:1 reagent ratio synthesis.

**Table 1 molecules-29-01686-t001:** The polyphenolic content in the herbal infusions used in the synthesis of SeNPs.

	Plant Material Used for SeNP Synthesis				
	Blackberry	Hop	Lemon Balm	Raspberry	Sage
**Flavonoids *****					
Kaempferol	<LOD	0.705 ± 0.023 ^a^	0.507 ± 0.020 ^b^	<LOD	0.748± 0.030 ^a^
Epicatechin	9.62 ± 0.324 ^a^	7.54 ± 0.295 ^b^	3.00 ± 0.101 ^c^	17.7 ± 0.730 ^d^	5.08 ± 0.220 ^e^
Catechin	3.91 ± 0.183 ^a^	1.22 ± 0.045 ^b^	0.11 ± 0.003 ^c^	2.02 ± 0.092 ^d^	0.10 ± 0.002 ^c^
EGCG	0.197 ± 0.007 ^a^	<LOD	<LOD	<LOD	0.180 ± 0.040 ^b^
Quercetin	0.064 ± 0.003 ^a^	0.608 ± 0.025 ^b^	0.078 ± 0.002 ^c^	0.077 ± 0.003 ^c^	0.052 ± 0.002 ^d^
Naringenin	<LOD	2.47 ± 0.100 ^a^	<LOD	0.237 ± 0.011 ^b^	0.312 ± 0.014 ^c^
Hesperetin	<LOD	8.18 ± 0.305 ^a^	<LOD	<LOD	<LOD
Myricetin	<LOD	<LOD	0.087 ± 0.003 ^a^	<LOD	0.161 ± 0.006 ^b^
Apigenin	0.09 ± 0.003 ^a^	<LOD	0.138 ± 0.004 ^b^	<LOD	1.05 ± 0.041 ^c^
Naringin	<LOD	<LOD	<LOD	<LOD	0.717± 0.024 ^a^
Luteolin	0.15 ± 0.006 ^a^	<LOD	<LOD	0.06 ± 0.002 ^b^	2.01 ± 0.093 ^c^
Rutin	1.45 ± 0.064 ^a^	<LOD	<LOD	2.49 ± 0.102 ^b^	0.347 ± 0.012 ^c^
**Polyphenolic acid *****					
Chlorogenic acid	7.58 ± 0.325 ^a^	2.93 ± 0.110 ^b^	0.217 ± 0.09 ^c^	3.71 ± 0.142 ^d^	0.458 ± 0.018 ^e^
pHBA	1.07 ± 0.045 ^a^	1.64 ± 0.076 ^b^	1.45 ± 0.032 ^c^	1.86 ± 0.057 ^d^	0.749 ± 0.08 ^e^
Caffeic acid	0.568 ± 0.021 ^a^	0.197 ± 0.007 ^b^	5.60 ± 0.19 ^c^	0.395 ± 0.013 ^d^	6.43 ± 0.304 ^e^
Ferulic acid	0.874 ± 0.039 ^a^	1.20 ± 0.041 ^b^	<LOD	0.619 ± 0.023 ^c^	2.24 ± 0.100 ^d^
Protocatechuic acid	0.605 ± 0.027 ^a^	2.22 ± 0.100 ^b^	<LOD	<LOD	1.09 ± 0.042 ^c^
p-coumaric acid	0.195 ± 0.007 ^a^	<LOD	<LOD	0.131 ± 0.003 ^b^	0.614 ± 0.0271 ^c^
Gallic acid	0.253 ± 0.010 ^a^	<LOD	<LOD	0.120 ± 0.003 ^b^	0.07 ± 0.002 ^c^

* Results are expressed in mg/L as the mean ± SD of three independent repetitions. Different letters in each row mean a difference at a significance level of *p* = 0.05. LOD—limit of detection, the lowest concentration of the polyphenolic compound that can be detected by the applied method. In our method, it is 0.01 mg/L.

**Table 2 molecules-29-01686-t002:** Antioxidant activity of herbal extracts and synthesized SeNPs. Abbreviations: the numbers indicate Se/herbal ratio, e.g., 11 means ratio 1:1, H-heating.

	FC [mgGa/L]	CUPRAC [mmolTr/L]	DPPH [mmolTr/L]	OH [%]	AOX
**Blackberry**					
Extract (1/1)	41.3 ± 1.83 ^a^	1.401 ± 0.05 ^a^	0.871 ± 0.02 ^a^	88.0 ± 3.21	97.0 ± 3.26
SeNPs11	32.4 ± 1.50 ^b^	1.368 ± 0.04 ^b^	0.748 ± 0.01 ^b^	99.0 ± 4.07 ^a^	98.5 ± 4.22 ^a^
SeNPs11H	24.1 ± 1.05 ^c^	1.359 ± 0.04 ^b^	0.741 ± 0.02 ^c^	61.2 ± 2.95 ^b^	98.2 ± 4.01 ^a^
Extract (1/2)	81.2 ± 3.82 ^d^	1.414 ± 0.03 ^c^	0.881 ± 0.03 ^d^	89.0 ± 3.84 ^c^	95.3 ± 4.11 ^b^
SeNPs12	60.3 ± 2.85 ^e^	1.405 ± 0.04 ^a^	0.753 ± 0.03 ^e^	93.7 ± 3.95 ^d^	96.9 ± 4.09 ^c^
SeNPs12H	52.2 ± 2.33 ^f^	1.417 ± 0.02 ^c^	0.751 ± 0.02 ^e^	89.9 ± 4.02 ^c^	95.7 ± 3.66 ^b^
Extract (1/3)	133.2 ± 5.23 ^g^	1.452 ± 0.03 ^d^	0.860 ± 0.03 ^f^	89.1 ± 3.75 ^c^	98.2 ± 3.96 ^a^
SeNPs13	103.8 ± 4.23 ^h^	1.394 ± 0.03 ^e^	0.748 ± 0.03 ^b^	84.7 ± 2.98 ^e^	99.4 ± 4.04 ^d^
SeNPs13H	93.5 ± 4.21 ^i^	1.388 ± 0.02 ^e^	0.757 ± 0.02 ^g^	70.5 ± 3.32 ^f^	99.0 ± 4.32 ^d^
**Hop**					
Extract (1/1)	9.06 ± 0.37 ^a^	1.22 ± 0.05 ^a^	0.455 ± 0.02 ^a^	53.3 ± 2.07 ^a^	97.0 ± 4.02 ^a^
SeNPs11	11.2 ± 0.48 ^b^	0.957 ± 0.03 ^b^	0.289 ± 0.01 ^b^	99.0 ± 4.11 ^b^	97.7 ± 3.75 ^a^
SeNPs11H	10.9 ± 0.45 ^b^	1.08 ± 0.02 ^c^	0.296 ± 0.02 ^c^	92.1 ± 3.25 ^c^	98.3 ± 3.98 ^b^
Extract (1/2)	17.0 ± 0.67 ^c^	1.40 ± 0.04 ^d^	0.685 ± 0.03 ^d^	70.0 ± 2.73 ^d^	98.2 ± 3.33 ^b^
SeNPs12	17.2 ± 0.73 ^c^	1.36 ± 0.03 ^e^	0.621 ± 0.02 ^e^	80.1 ± 3.35 ^e^	99.4 ± 4.00 ^c^
SeNPs12H	17.8 ± 0.65 ^c^	1.39 ± 0.05 ^e^	0.666 ± 0.02 ^f^	77.6 ± 2.87 ^f^	96.6 ± 3.22 ^d^
Extract (1/3)	26.2 ± 1.22 ^d^	1.47 ± 0.06 ^f^	0.837 ± 0.03 ^g^	84.9 ± 3.82 ^g^	97.4 ± 3.73 ^a^
SeNPs13	17.3 ± 0.74 ^c^	1.41 ± 0.03 ^d^	0.749 ± 0.03 ^h^	77.8 ± 3.70 ^f^	99.0 ± 3.54 ^c^
SeNPs13H	22.8 ± 1.03 ^e^	1.42 ± 0.04 ^d^	0.747 ± 0.02 ^h^	75.7 ± 2.92 ^h^	96.7 ± 2.91 ^d^
**Lemon balm**					
Extract (1/1)	119.9 ± 4.56 ^a^	1.42 ± 0.06 ^a^	0.872 ± 0.04 ^a^	96.1 ± 3.78 ^a^	97.3 ± 3.47 ^a^
SeNPs11	98.5 ± 3.47 ^b^	0.840 ± 0.03 ^b^	0.810 ± 0.03 ^b^	98.0 ± 2.36 ^b^	99.6 ± 3.29 ^b^
SeNPs11H	98.9 ± 2.36 ^b^	0.958 ± 0.04 ^c^	0.824 ± 0.03 ^c^	99.0 ± 3.33 ^c^	96.6 ± 2.41 ^c^
Extract (1/2)	196.9 ± 8.27 ^c^	1.44 ± 0.05 ^a^	0.845 ± 0.04 ^d^	99.6 ± 4.05 ^c^	98.0 ± 3.20 ^d^
SeNPs12	212.9 ± 8.51 ^d^	1.32 ± 0.04 ^d^	0.791 ± 0.02 ^e^	99.5 ± 3.95 ^c^	99.2 ± 4.08 ^b^
SeNPs12H	215.6 ± 9.02 ^d^	1.41 ± 0.03 ^a^	0.824 ± 0.03 ^c^	99.0 ± 2.98 ^c^	98.2 ± 3.21 ^d^
Extract (1/3)	215.8 ± 7.30 ^d^	0.999 ± 0.02	0.839 ± 0.03 ^f^	97.0 ± 2.33 ^d^	98.7 ± 4.11 ^d^
SeNPs13	100.9 ± 4.08 ^b^	1.43 ± 0.02 ^a^	0.790 ± 0.02 ^e^	83.1 ± 3.07 ^e^	99.1 ± 3.79 ^b^
SeNPs13H	185.7 ± 3.33 ^e^	1.41 ± 0.03 ^a^	0.765 ± 0.02 ^g^	92.5 ± 4.04 ^f^	99.1 ± 3.84 ^b^
**Raspberry**					
Extract (1/1)	26.8 ± 1.12 ^a^	1.300 ± 0.05 ^a^	0.871 ± 0.04 ^a^	26.2 ± 1.10 ^a^	97.3 ± 3.60 ^a^
SeNPs11	15.4 ± 0.63 ^b^	1.390 ± 0.06 ^b^	0.392 ± 0.01 ^b^	89.2 ± 3.23 ^b^	98.3 ± 4.15 ^b^
SeNPs11H	14.4 ± 0.51 ^c^	1.337 ± 0.03 ^c^	0.492 ± 0.02 ^c^	86.9 ± 4.20 ^c^	96.0 ± 3.89 ^c^
Extract (1/2)	58.5 ± 2.37 ^d^	1.067 ± 0.02 ^d^	0.867 ± 0.03 ^a^	30.1 ± 0.957 ^d^	98.0 ± 4.08 ^b^
SeNPs12	36.9 ± 1.60 ^d^	1.372 ± 0.04 ^e^	0.854 ± 0.04 ^d^	98.2 ± 4.32 ^e^	94.9 ± 3.54 ^d^
SeNPs12H	37.6 ± 1.73 ^d^	1.366 ± 0.03 ^f^	0.773 ± 0.02 ^e^	81.6 ± 3.70 ^f^	99.3 ± 4.24 ^e^
Extract (1/3)	84.2 ± 3.20 ^e^	1.313 ± 0.05 ^g^	0.854 ± 0.02 ^d^	32.4 ± 1.52 ^d^	98.7 ± 3.89 ^b^
SeNPs13	72.0 ± 2.37 ^f^	1.064 ± 0.02 ^h^	0.798 ± 0.02 ^f^	99.0 ± 4.33 ^e^	98.5 ± 3.47 ^b^
SeNPs13H	72.1 ± 2.75 ^f^	1.384 ± 0.04 ^b^	0.746 ± 0.01 ^g^	30.7 ± 0.992 ^d^	99.0 ± 4.18 ^e^
**Sage**					
Extract (1/1)	32.1 ± 1.22 ^a^	1.414 ± 0.05 ^a^	0.859 ± 0.03 ^a^	30.4 ± 1.03 ^a^	97.6 ± 3.87 ^a^
SeNPs11	19.2 ± 0.831 ^b^	1.284 ± 0.05 ^b^	0.845 ± 0.04 ^b^	90.3 ± 3.98 ^b^	99.5 ± 3.87 ^b^
SeNPs11H	18.7 ± 0.673 ^b^	1.361 ± 0.04 ^c^	0.725 ± 0.02 ^c^	91.7 ± 4.07 ^b^	97.4 ± 2.53 ^a^
Extract (1/2)	67.7 ± 2.31 ^c^	1.394 ± 0.03 ^d^	0.731 ± 0.03 ^c^	34.2 ± 1.63 ^c^	97.9 ± 3.75 ^a^
SeNPs12	46.9 ± 1.83 ^d^	1.388 ± 0.03 ^e^	0.740 ± 0.03 ^d^	99.2 ± 3.87 ^d^	98.7 ± 4.08 ^c^
SeNPs12H	53.6 ± 2.31 ^e^	1.400 ± 0.05 ^d^	0.745 ± 0.02 ^d^	99.0 ± 3.32 ^d^	95.3 ± 3.77 ^d^
Extract (1/3)	94.4 ± 3.81 ^f^	1.426 ± 0.06 ^f^	0.827 ± 0.04 ^e^	24.1 ± 1.01 ^e^	98.2 ± 4.05 ^c^
SeNPs13	93.3 ± 4.08 ^f^	1.070 ± 0.04 ^g^	0.770 ± 0.03 ^f^	99.0 ± 3.76 ^d^	98.8 ± 4.20 ^c^
SeNPs13H	76.6 ± 2.93 ^g^	1.400 ± 0.05 ^d^	0.750 ± 0.02 ^g^	99.0 ± 2.98 ^d^	97.6 ± 3.88 ^a^

Results are expressed as the mean ± SD of three independent repetitions. Different letters in each section for a specific herbal extract mean a difference at a significance level of *p* = 0.05.

**Table 3 molecules-29-01686-t003:** Antibacterial activity (MIC) of herbal extracts and synthesized SeNPs. Abbreviations: the numbers indicate Se/herbal ratio, e.g., 11 means ratio 1:1, H-heating, NoI—no inhibition.

	MIC (%)			MIC (%)	
	*Escherichia coli*	*Staphylococcus aureus*		*Escherichia coli*	*Staphylococcus aureus*
**Blackberry**			**Lemon balm**		
Extract (1/1)	100	25	Extract (1/1)	NoI	NoI
SeNPs11	25	12.5	SeNPs11	100	50
SeNPs11H	25	12.5	SeNPs11H	100	50
Extract (1/2)	100	12.5	Extract (1/2)	NoI	NoI
SeNPs12	50	3	SeNPs12	100	100
SeNPs12H	50	3	SeNPs12H	100	100
Extract (1/3)	50	12.5	Extract (1/3)	NoI	NoI
SeNPs13	25	6	SeNPs13	NoI	100
SeNPs13H	25	6	SeNPs13H	NoI	100
**Hop**			**Raspberry**		
Extract (1/1)	NoI	NoI	Extract (1/1)	50	50
SeNPs11	50	12.5	SeNPs11	25	25
SeNPs11H	50	12.5	SeNPs11H	50	50
Extract (1/2)	NoI	NoI	Extract (1/2)	100	50
SeNPs12	100	6	SeNPs12	12.5	12.5
SeNPs12H	100	6	SeNPs12H	12.5	25
Extract (1/3)	NoI	NoI	Extract (1/3)	100	100
SeNPs13	NoI	6	SeNPs13	12.5	12.5
SeNPs13H	NoI	6	SeNPs13H	12.5	6
**Sage**			**Chemically synthesized SeNPs**		
Extract (1/1)	100	50	SeNPs AA	100	6
SeNPs11	25	25	SeNPs AApvA	100	6
SeNPs11H	50	25	SeNPs GA	25	1.6
Extract (1/2)	50	50	SeNPs GApvA	25	1.6
SeNPs12	12.5	12.5			
SeNPs12H	12.5	12.5			
Extract (1/3)	50	50			
SeNPs13	12.5	6			
SeNPs13H	12.5	6			

## Data Availability

The data presented in this study are available upon request from the corresponding author. Raw data will be deposited at the University of Warsaw Research Data Repository. The base is available at https://danebadawcze.uw.edu.pl// (accessed on 1 January 2024).

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
