# Peer review of "Herbal Polyphenols as Selenium Reducers in the Green Synthesis of Selenium Nanoparticles: Antibacterial and Antioxidant Capabilities of the Obtained SeNPs"

_molecules, 2024, doi:10.3390/molecules29081686_

Round 1

Reviewer 1 Report

Comments and Suggestions for Authors

First and foremost, I would like to thank the authors for submitting this research article, which explores the green synthesis of selenium nanoparticles (SeNPs) using various herbal extracts and evaluates their antioxidant and antibacterial properties. This is an intriguing and significant area of research with potential contributions to the fields of nanotechnology and biomedical applications. After careful review and consideration, I believe the article holds scientific value and practical prospects. However, to further enhance the quality and readability of the article, I recommend some minor revisions before acceptance.

1. Methodological Clarity

I suggest that the authors provide more detailed descriptions of the experimental methods, including the specific preparation processes for the herbal extracts, detailed conditions for nanoparticle synthesis (such as temperature, duration, solvents, etc.), and the detailed steps and standards used for evaluating antioxidant and antibacterial properties. This would help other researchers replicate the experiments, enhancing the transparency and credibility of the research.

2. Data Completeness

Where possible, it would be beneficial for the authors to provide more raw data and supplementary materials, such as high-resolution images from SEM and TEM, DLS data charts, and detailed data from antioxidant and antibacterial tests. This would enable readers to better understand the research findings.

3. Results Interpretation and Discussion

While the article already discusses the experimental results to some extent, I encourage the authors to further delve into the mechanisms behind the antioxidant and antibacterial properties of SeNPs and discuss which specific components of the herbal extracts might influence these properties. Additionally, the discussion section could benefit from a comparison with existing literature, highlighting the novelty and potential practical applications of this research.

4. Structure and Language

The authors are advised to review and improve the language and structure of the article to enhance its clarity and fluency. Especially in the introduction and discussion sections, ensure logical coherence and avoid unnecessary repetition. Consider hiring a professional scientific editor or a native English-speaking colleague for language proofreading if possible.

5. Literature Update

Lastly, I recommend that the authors update and supplement the references with the latest relevant literature to ensure the comprehensiveness and timeliness of the literature review, further solidifying the theoretical foundation and context of the research.

In summary, the article provides valuable insights and demonstrates the potential of herbal extract-mediated synthesis of SeNPs. I believe that by addressing the suggestions above, the manuscript can be significantly improved, making it more appealing to researchers and practitioners in the field. I look forward to seeing the revised manuscript.

Author Response

At the beginning, we would like to thank you for your thorough review and comments. We have done our best to meet your expectations. Details of the corrections that we made are described below.

Comments and Suggestions for Authors

First and foremost, I would like to thank the authors for submitting this research article, which explores the green synthesis of selenium nanoparticles (SeNPs) using various herbal extracts and evaluates their antioxidant and antibacterial properties. This is an intriguing and significant area of research with potential contributions to the fields of nanotechnology and biomedical applications. After careful review and consideration, I believe the article holds scientific value and practical prospects. However, to further enhance the quality and readability of the article, I recommend some minor revisions before acceptance.

1. Methodological Clarity

I suggest that the authors provide more detailed descriptions of the experimental methods, including the specific preparation processes for the herbal extracts, detailed conditions for nanoparticle synthesis (such as temperature, duration, solvents, etc.), and the detailed steps and standards used for evaluating antioxidant and antibacterial properties. This would help other researchers replicate the experiments, enhancing the transparency and credibility of the research.

All the details of the procedures used have been included in the manuscript in section 4.

  1. Data Completeness

    Where possible, it would be beneficial for the authors to provide more raw data and supplementary materials, such as high-resolution images from SEM and TEM, DLS data charts, and detailed data from antioxidant and antibacterial tests. This would enable readers to better understand the research findings.

Raw data will be deposited at the University of Warsaw Research Data Repository. The base is available at https://danebadawcze.uw.edu.pl//. This information was added to the manuscript in the 763 line.

  1. Results Interpretation and Discussion

    While the article already discusses the experimental results to some extent, I encourage the authors to further delve into the mechanisms behind the antioxidant and antibacterial properties of SeNPs and discuss which specific components of the herbal extracts might influence these properties. Additionally, the discussion section could benefit from a comparison with existing literature, highlighting the novelty and potential practical applications of this research.

The discussion and results interpretation were extended.

4. Structure and Language

The authors are advised to review and improve the language and structure of the article to enhance its clarity and fluency. Especially in the introduction and discussion sections, ensure logical coherence and avoid unnecessary repetition. Consider hiring a professional scientific editor or a native English-speaking colleague for language proofreading if possible.

The manuscript has been improved in terms of language. In addition, we will use the language correction option offered to the accepted publications by the Publisher.

5. Literature Update

Lastly, I recommend that the authors update and supplement the references with the latest relevant literature to ensure the comprehensiveness and timeliness of the literature review, further solidifying the theoretical foundation and context of the research.

 Several new literature references were cited and the research described in them was discussed in the manuscript.

In summary, the article provides valuable insights and demonstrates the potential of herbal extract-mediated synthesis of SeNPs. I believe that by addressing the suggestions above, the manuscript can be significantly improved, making it more appealing to researchers and practitioners in the field. I look forward to seeing the revised manuscript.

Reviewer 2 Report

Comments and Suggestions for Authors

The significance of selenium as an indispensable trace element for human health has been increasingly recognized, particularly in the context of its nanoparticulate form, selenium nanoparticles (SeNPs), which exhibit promising potential for medical applications. This study explores the green synthesis of SeNPs utilizing herbal extracts, focusing on the impact of various factors such as the choice of herbal species, reagent ratios, and the application of post-reaction heating on the antibacterial and antioxidant effectiveness of the resulting SeNPs. Additionally, the investigation extends to the correlation between these biological activities and the physical attributes of the SeNPs, including size and shape. Findings from the study clearly demonstrate that SeNPs synthesized through this method display significantly enhanced antioxidant and antibacterial properties when compared to the herbal extracts used in their formation. Notably, the process of heating the post-reaction mixture was found to have no detrimental impact on the size, shape, or other relevant properties of the SeNPs. This research underscores the high quality and efficacy of SeNPs produced via the green synthesis approach, leveraging herbal polyphenols, and highlights their potential for broader application in medical and health-related fields.

Comments on the Quality of English Language

Minor editing of English language required

Author Response

The significance of selenium as an indispensable trace element for human health has been increasingly recognized, particularly in the context of its nanoparticulate form, selenium nanoparticles (SeNPs), which exhibit promising potential for medical applications. This study explores the green synthesis of SeNPs utilizing herbal extracts, focusing on the impact of various factors such as the choice of herbal species, reagent ratios, and the application of post-reaction heating on the antibacterial and antioxidant effectiveness of the resulting SeNPs. Additionally, the investigation extends to the correlation between these biological activities and the physical attributes of the SeNPs, including size and shape. Findings from the study clearly demonstrate that SeNPs synthesized through this method display significantly enhanced antioxidant and antibacterial properties when compared to the herbal extracts used in their formation. Notably, the process of heating the post-reaction mixture was found to have no detrimental impact on the size, shape, or other relevant properties of the SeNPs. This research underscores the high quality and efficacy of SeNPs produced via the green synthesis approach, leveraging herbal polyphenols, and highlights their potential for broader application in medical and health-related fields.

Comments on the Quality of English Language

Minor editing of English language required

The manuscript has been improved in terms of language. In addition, we will use the language correction option offered to the accepted publications by the Publisher.

Reviewer 3 Report

Comments and Suggestions for Authors

The authors present a study on the preparation of selenium nanoparticles by reduction of SeO2 with plant extracts. The study includes the preparation of nanoparticles and their characterisation (SEM, TEM). The plant extracts used in the study were analysed by HPLC to determine the content of natural products (mainly polyphenols) present in the biological material. The resulting nanoparticles were tested for antioxidant and antimicrobial activity and their effect on bacterial biofilm formation was determined.

The studies were carried out in a careful and methodologically correct manner. An interesting result is the determination of the biological dependence of the nanoparticles obtained on the extract used. Nanoparticles obtained by reduction with pure ascorbic acid also differed significantly from the materials described in the publication. The authors found no correlation between the physical properties of the nanoparticles and their antioxidant and antimicrobial activity. This result is interpreted as an influence of the bioactive components of the extract. This research is a continuation of the numerous scientific works discussed in the review article {https://doi.org/10.1088/2053-1591/ab29d6}. In my opinion, the discussion in the publication needs to be expanded.

1.The authors focus on the effect of the extracts on the SeO2 reduction process. The effect of SeO2 on the biologically active components of the matrix is completely ignored. The reactivity of this reagent leading to the formation of various potentially biologically active substances, including seleno-organic compounds, is well known (e.g. DOI:10.1021/acs.joc.8b00558.) These reactions are typical of carbonyl compounds that may be present in plant extracts. The diversity of nanoparticles and the lack of correlation of their activity with physicochemical properties suggest that their properties may be influenced (as suggested by the authors) by matrix-derived substances adsorbed or chemically bound to the nanoparticle surface, but also by the products of their reaction with SeO2. 

2. Additional justification is also required for the design of the work itself.  If the purpose of the research is purely exploratory - it would be more efficient to use pure natural products or mixtures of defined natural products for reduction. On the other hand, if the research is intended to lead to pharmaceutical preparations, the use of crude extracts is of course economically justified, but how do the authors envisage registering a potent product (selenium has a narrow range between therapeutic and toxic dose) with an undefined composition depending on the variable composition of the plant extract?

Author Response

The authors present a study on the preparation of selenium nanoparticles by reduction of SeO2 with plant extracts. The study includes the preparation of nanoparticles and their characterisation (SEM, TEM). The plant extracts used in the study were analysed by HPLC to determine the content of natural products (mainly polyphenols) present in the biological material. The resulting nanoparticles were tested for antioxidant and antimicrobial activity and their effect on bacterial biofilm formation was determined.

The studies were carried out in a careful and methodologically correct manner. An interesting result is the determination of the biological dependence of the nanoparticles obtained on the extract used. Nanoparticles obtained by reduction with pure ascorbic acid also differed significantly from the materials described in the publication. The authors found no correlation between the physical properties of the nanoparticles and their antioxidant and antimicrobial activity. This result is interpreted as an influence of the bioactive components of the extract. This research is a continuation of the numerous scientific works discussed in the review article {https://doi.org/10.1088/2053-1591/ab29d6}. In my opinion, the discussion in the publication needs to be expanded.

The discussion and results interpretation were extended.

1.The authors focus on the effect of the extracts on the SeO2 reduction process. The effect of SeO2 on the biologically active components of the matrix is completely ignored. The reactivity of this reagent leading to the formation of various potentially biologically active substances, including seleno-organic compounds, is well known (e.g. DOI:10.1021/acs.joc.8b00558.) These reactions are typical of carbonyl compounds that may be present in plant extracts. The diversity of nanoparticles and the lack of correlation of their activity with physicochemical properties suggest that their properties may be influenced (as suggested by the authors) by matrix-derived substances adsorbed or chemically bound to the nanoparticle surface, but also by the products of their reaction with SeO2. 

 It was added to the text.  

2. Additional justification is also required for the design of the work itself.  If the purpose of the research is purely exploratory - it would be more efficient to use pure natural products or mixtures of defined natural products for reduction. On the other hand, if the research is intended to lead to pharmaceutical preparations, the use of crude extracts is of course economically justified, but how do the authors envisage registering a potent product (selenium has a narrow range between therapeutic and toxic dose) with an undefined composition depending on the variable composition of the plant extract?

The justification for conducting the described research has been added to the conclusions section.